# The Functional Interplay between Gut Microbiota, Protein Hydrolysates/Bioactive Peptides, and Obesity: A Critical Review on the Study Advances

**DOI:** 10.3390/antiox11020333

**Published:** 2022-02-08

**Authors:** Simon Okomo Aloo, Deog-Hwan Oh

**Affiliations:** Department of Food Science and Biotechnology, College of Agriculture and Life Sciences, Kangwon National University, Chuncheon 24341, Gangwon-do, Korea; okomosimon@gmail.com

**Keywords:** anti-obesity, a bioactive peptide, gut microbiota, interaction, protein hydrolysate

## Abstract

Diet is an essential factor determining the ratio of pathogenic and beneficial gut microbiota. Hydrolysates and bioactive peptides have been described as crucial ingredients from food protein that potentially impact human health beyond their roles as nutrients. These compounds can exert benefits in the body, including modulation of the gut microbiota, and thus, they can reduce metabolic disorders. This review summarized studies on the interaction between hydrolysates/peptides, gut microbes, and obesity, focusing on how hydrolysates/peptides influence gut microbiota composition and function that improve body weight. Findings revealed that gut microbes could exert anti-obesity effects by controlling the host’s energy balance and food intake. They also exhibit activity against obesity-induced inflammation by changing the expression of inflammatory-related transcription factors. Protein hydrolysates/peptides can suppress the growth of pro-obesity gut bacteria but facilitate the proliferation of those with anti-obesity effects. The compounds provide growth factors to the beneficial gut bacteria and also improve their resistance against extreme pH. Hydrolysates/peptides are good candidates to target obesity and obesity-related complications. Thus, they can allow the development of novel strategies to fight incidences of obesity. Future studies are needed to understand absorption fate, utilization by gut microbes, and stability of hydrolysates/peptides in the gut under obesity.

## 1. Introduction

The prevalence of obesity is rising worldwide contributing to a high risk of developing numerous related metabolic complications, such as cardiovascular and diabetes type 2 diseases. Several factors may directly or indirectly lead to obesity. Studies have associated lifestyle, genetics, and environmental-related factors as the significant causes of obesity [1]. The human gut contains millions of microbes. The number of microorganisms in the gut exceeds 10^14^, which means there are ten times more bacterial cells than human cells [2]. While some of these organisms are known for their potential roles in regulating various body physiological processes that positively impact human health, others are considered pathogenic microorganisms capable of triggering illnesses [2]. Thus, an increasing body of evidence suggests that gut microbiota homeostasis plays a key role in the pathogenesis of obesity and obesity-related complications. The role of gut microbiota in obesity has been reviewed elsewhere [3,4,5]. Dietary and nutritional modulation can significantly change gut microbiota composition by either increasing the abundance of beneficial microbes, such as *Lactobacillus* and *Bifidobacterium* or decreasing the number of opportunistic pathogens, such as the *Enterobacteriaceae, Desulfovibrionaceae*, and *Streptococcaceae* families [5].

The use of ingredients derived from food protein in industries has been in existence for decades, and the basic manufacturing process of most food-derived protein products has remained the same. A great diversity of biologically active protein by-products has been produced from protein-rich foods, such as milk proteins, rice soy proteins, and other food sources [6]. Efficient and reproducible biocatalytic technologies, such as enzymatic digestion, gastrointestinal digestion, and microbial fermentation via in vitro or in vivo methods can produce valuable hydrolyzed products from food-based sources that may help reduce excess weight gain and other physiological disorders [6]. Some of these products are bioactive peptides or hydrolysates, which have great biological functions in the body. Food hydrolysates/bioactive peptides have been shown to act as anti-inflammatories, antioxidants, anti-diabetic, anticancer, anti-obesity, and antimicrobial agents, among many other health-related functions [7]. Currently, bioactive peptides and other forms of food hydrolysates have attracted attention in treating metabolic disorders, and their functions as anti-obesity compounds have been investigated and reviewed [7,8].

Scientific research indicates that food hydrolysates/bioactive peptides derived from various diets can promote the growth of probiotics and reduce the risks of metabolic disorders [9]. Albumin-derived hydrolysates [1], soy bioactive peptides [10], and pepsin egg white hydrolysates [11] have been confirmed to significantly contribute to gut microbiota composition and function, which affects body weight. Nevertheless, despite their potential role in preventing excess weight gain, only a limited number of studies have reported on the impacts of food hydrolysates on gut microbiota composition, which affects body weight. The topic of interactions between food hydrolysates/peptides, gut bacteria, and obesity has rarely been addressed in the literature except for the recently published book chapter on the bioactive peptides against inflammatory intestinal disorders and obesity (pp. 155–183) [12]. In this review article, the interaction between protein hydrolysates, gut microbiota, and body weight has been reviewed. First, we described the significance of gut microbiota composition and function on body weight, including their mechanisms of action. Subsequently, the preventive effects of food hydrolysates/bioactive peptides on obesity and related complications by modifying the gut microbial composition and function and the mechanism of action involved have been discussed. The article further presents current issues arising from marketing bioactive peptides, legal requirements, as well as providing some areas that require investigations for future studies. Thus, this review offers the latest insight on the physiological effects of the host’s gut microbiota on obesity and the importance of food hydrolysates/bioactive peptides in preventing obesity by regulating the gut’s microbial composition and functions.

## 2. Gut Microbiota and Obesity

### 2.1. Gut Microbial Composition and the Significance in Obesity

Gut microbiota are the microorganisms that live in the digestive tracts of living organisms including humans and insects. Generally, about five major divisions of bacteria are present in the gut: *Firmicutes*, *Actinobacteria*, *Fusobacteria*, *Proteobacteria*, and *Bacteroidetes*. However, in humans, *Firmicutes* and *Bacteroidetes* are more numerous than the remaining three groups. The other microbes present in the gut include fungi (*Candida*, *Penicillium*, *Saccharomyces*, and *Aspergillus*) and *Archaea* species [10]. The microbial composition along the gut may change depending on the acidity and oxygen levels, which determine their distribution. The proximal gastrointestinal tract is dominated by *Firmicutes, Lactobacilli*, and *Proteobacteria*, whereas the distal gastrointestinal tract contains mainly *Bacteroidetes, Firmicutes*, and *Akkermansia muciniphila* [13]. Recent reports suggest that the nature and composition of the intestinal microbiota can significantly be altered in obesity [14]. The individuals with lean body mass are said to have more *Bacteroidetes*, whereas those who are obese have more *Firmicutes* clusters in their gut [14]. Thus, studies have generally agreed that gut microbiota dysbiosis affects the ability of the host to extract energy from ingested food and utilize or store this energy in the adipose tissues, thereby influencing an individual’s body weight [15]. Requena et al. studied this matter to determine if there is a deviation between the microbiological community of the obese and lean Zucker rats [11]. The result showed that obese and lean rats differed significantly in several microbiological parameters [11]. Counts per gram of feces of total bacteria indicated that *Lactobacillus/Enterococcus, C. leptum, Roseburia*, and *Ruminococcus* were significantly higher in the obese rats than in lean rats [11]. Thus, in the gut, the microbial composition can be widely grouped into two: beneficial (anti-obesity) gut microbiota reducing the risks of obesity and pathogenic (pro-obesity) gut microbiota increasing the risks of obesity. Therefore, due to the close relationship between obesity and intestinal bacteria, gut microbial composition plays a critical role in the pathogenesis of obesity. The pro-obesity and anti-obesity gut microbiota groups have been discussed in the subsequent sections.

### 2.2. Pro-Obesity Gut Microbiome

Most of the incidences of obesity are attributed to diet, and since diet can modify the composition of gut microbiota, recent efforts to combat obesity have focused on the roles of the gut microbiota as a mediator that can predispose the host to diet-induced excess weight gain. In this viewpoint, attempts have been directed towards identifying key obesity-related microorganisms in the gut which can potentially affect the pathogenesis of obesity by either inducing weight gain or weight loss. Evidence indicates that there is a dysbiosis of gut microbiota composition in obesity. As stated earlier, studies on gut microbiota suggest that most of the bacteria present in the gut and feces belong to *Bacteroidetes* and *Firmicutes*. *Firmicutes* are a phylum of bacteria, most of which have Gram-positive cell wall structures. In the gut, *Firmicutes* is composed of *Lactobacillus, Ruminococcus, Peptococcus, Clostridium, Eubacterium, Faecalibacterium*, and *Peptostreptococcus* [10]. A higher *Firmicutes/Bacteroidetes* ratio is associated with increased obesity in most animal study models and has long been used as a physiological marker for obesity [11]. However, nowadays, instead of the whole *Firmicutes* phylum being associated with obesity, research has been designed to target genera or species within this phylum that are more likely to cause obesity. This is especially due to the fact that some of the bacteria from *Firmicutes* phylum possess protective effects on weight gain instead of being pathogenic. For instance, *Lactobacillus* is a bacterial genus belonging to *Firmicutes* phylum. Despite being associated with obesity, some species of *Lactobacillus*, such as *Lactobacillus paracasei* and *Lactobacillus plantarum* have protective effects against weight gain [16]. On the other hand, *Erysipelotrichia (Mollicutes)* is a class of bacteria thought to be pro-obesity within *Firmicutes* phylum [17]. The species from the class *Mollicutes (Erysipelotrichia)* is known to induce obesity via an inflammation-dependent pathway [17].

In other findings, studies have also described lipopolysaccharide-producing bacteria as inducers of excess weight gain. Families in the phylum *Proteobacteria* including *Enterobacteriaceae* and *Desulfovibrionaceae*, contain many pathogens capable of producing lipopolysaccharide as an endotoxin [18]. The composition of these organisms has been shown to increase in obese mice compared to lean mice [18]. Other endotoxin-producing bacteria, such as *Enterobacter* have also been connected with high risks of obesity [19]. Fei et al. established that *Enterobacter* isolated from the obese human gut could trigger weight gain and insulin resistance in germ-free mice [19]. The study also showed that *Enterobacter* induced not only obesity but also triggered obesity-related pro-inflammatory responses in the mice [19]. Moreover, Liu et al. found that potential opportunistic pathogens, such as *Ruminococcus torques, Ruminococcus gnavus, Dorea longicatena, Doressa formicigenerans*, and *Coprococcus comes*, and a cluster consisting of *Lachnospiraceae bacterium, Fusobacterium ulcerans*, and *Fusobacterium varium*, were abundant in obese mice, indicating that these organisms are associated with increased risks of weight gain [20]. Other pathogenic bacteria, such as *Staphylococcus aureus* have also been implicated in obesity development in children and pregnant women [21,22].

### 2.3. Anti-Obesity Gut Microbiome

Anti-obesity gut microbiota diminishes the chances of obesity development by modulating physiological pathways involved in obesity pathogenesis. In most of the scientific findings, microbial bacteria belonging to phylum *Bacteroidetes* have been chiefly associated with anti-obesity effects. A study by Santacruz et al. found out that *Bacteroidetes*, such as *Bacteroides fragilis* were significantly higher following diet intervention and was correlated with weight loss in adolescents [23]. They also discovered that bacteria from the genus, *Bifidobacterium* including *Bifidobacterium catenulatum**, Bifidobacterium breve*, and *Bifidobacterium bifidum* were significantly increased in both high and low weight loss groups indicating that they play roles in restoring normal body weight [23]. It was also discovered that people with lean body mass have a high population of *Akkermansia muciniphila, Faecalibacterium, Methanobrevibacter smithii, Lactobacillus plantarum*, and *paracasei* [16]. Bacteria, such as *Faecalibacterium prausnitzii* is known to produce end products of fermentation, such as butyrate with anti-inflammatory and protective effects against obesity [16], while others, such as *Akkermansia muciniphila*, also known as *Verrucomicrobia* are involved in degrading and stimulating mucin in the host’s gut, controlling mucus, maintaining the integrity of the gut barrier, and promoting an optimal balance of gut microflora (eubiosis) in the host [16].

### 2.4. Contrasting Findings on Gut Microbiota and Obesity

Despite the evidence distinguishing pro and anti-obesity gut microbiota, there are cases where studies report conflicting findings on specific gut microbiota and its relation to body weight. One of such reports is on *Lactobacilli*. In some conclusions, studies reported that an increased *Lactobacilli* population is associated with risks of obesity [24,25]. In contrast, as reviewed in another article, some studies have demonstrated the anti-obesity effects of *Lactobacilli* [26]. Similarly, there is also an inconsistency in *F. prausnitzii* reports related to obesity. Balamurugan et al. described an increased *F. prausnitzii* population in obese patients [27] while other studies report contrasting findings [28]. In general, the above findings indicate that the gut microbiota has a crucial role in the pathophysiology of obesity although, it is still not very clear which microbes are indeed pro-obesity and which ones are anti-obesity due to the conflicting findings. Therefore, this area requires further scrutiny to distinguish the pathogenic and beneficial bacteria.

### 2.5. Modulating Effects of Gut Microbiota on Obesity: Mechanism Involved

The weight reduction mechanism by anti-obesity gut microbiota has not been fully explored and understood. However, studies have revealed an association between appetite control, energy balance, and microbiota. In humans, appetite is controlled by various factors, which are integrated into the central nervous system (CNS) [29]. Appetite-related neural signals are derived by the vagus nerve from the distension of the digestive system walls. Hormones involved in appetite control include leptin, cholecystokinin, insulin, cortisol, ghrelin, glucagon-like peptide 1, and peptide YY. Distension of the gut consists of the vagus nerve afferent, which sends signals to the brain and stimulates the release of these hormones from the gut mucosa, a process that is only possible via the gut–brain axis. The gut microbiota is involved in the regulation of food intake by influencing the release of hormones responsible for the food intake and through controlling transcription factors responsible for body energy balance via the gut–brain axis [29,30]. The gut microbiota can achieve this through various means. First, gut microbiota influences host energy balance through sensors of microbial products. Short-chain fatty acids are a sub-group of fatty acids composed of acetate, propionate, and butyrate, which are end products of bacterial fermentation. These metabolites may act as signaling molecules that regulate various transcription factors involved in energy balance [30]. The short-chain fatty acids and conjugated fatty acids can modulate the brain via direct or indirect mechanisms and affects its ability to regulate appetite and food intake [30].

Moreover, the short-chain fatty acids can bind to receptors on enteroendocrine cells, changing their ability to release hormones related to appetite into the systemic circulation, thereby modifying food intake [29]. Acetate, for example can directly suppress appetite via central hypothalamic mechanisms [30]. Kimura et al., in their study, demonstrated that gut microbiota influenced the development of obesity by suppressing insulin-mediated fat accumulation via the short-chain fatty acid receptor (majorly, acetate receptors) [31]. Gut microbiota, such as *Bifidobacterium* reduced fat accumulation in obese mice via activated short-chain fatty acid receptor GPR43 [32]. Therefore, gut microbiota metabolites are essential regulators of body weight and may significantly help regulate host lipid accumulation and food intake, all of which have essential implications on energy homeostasis and obesity. In addition to short-chain fatty acids, some gut bacteria, such as *Bifidobacterium adolescentis* are known to produce neuroactive metabolites, such as gamma-aminobutyric acid (GABA), catecholamines, and acetylcholine which may significantly suppress weight gain through means other than controlling gut hormones [33,34,35]. For instance, while serotonin may exert its appetite-suppressant potential by its modulating effect on lanocortin neurons [30], GABA is reported to reduce obesity-induced adipose tissue macrophage infiltration in adipose tissues [36]. Finally, in the human gut, it has been demonstrated that there exists an interaction between pro-obesity gut microbiota and anti-obesity gut microbiota through their metabolites, and this association has an important influence on obesity. For example, it was revealed that breastfeeding children acquire a high amount of *Bifidobacterium* in their early days of life [37]. The high profile of *Bifidobacterium* produces a high amount of acetate, which suppresses the growth of pro-obesity gut microbial bacteria, such as *Escherichia coli* and *Clostridium perfringens* [37]. Therefore, the characteristic gut microbiota and their metabolites, as well as their relationship may play a crucial role in the pathogenesis of obesity.

On the other hand, pro-obesity microbiomes have been reported to be involved in various activities that promote body weight gain. Some microbes, such as those belonging to the phylum *Firmicutes* are involved in promoting adiposity or could enhance host-mediated adaptive response mechanisms that limit energy uptake, such as reducing the capacity to ferment polysaccharides [38]. *Firmicutes* have been described to possess many carbohydrate metabolism enzymes, which can contribute to the metabolization of carbohydrates allowing a greater energy absorption and contributing to obesity [16]. Furthermore, the pro-obesity gut microbiota is involved in inducing low-grade inflammation by promoting metabolic expression of inflammatory markers in adipose tissue and pro-inflammatory cytokines associated with increased risks of weight gain [39]. It was discovered that gut microbiota-associated inflammation is controlled by microbiota lipopolysaccharide (LPS) [40]. The lipopolysaccharide from intestinal bacteria may increase the risk of developing obesity and cause insulin resistance via means including inducing obesity-inflammatory markers in adipose tissue [37,39]. Caesar et al. found out that gut microbiota-derived lipopolysaccharide, such as that released by *E. coli* is actively involved in the accumulation of pro-inflammatory factors in white adipose tissue (WAT) [39]. In the study, Caesar et al. discovered that colonization with lipopolysaccharide-producing *E. coli* promoted increased susceptibility to obesity by promoting adiposity in mice [39]. Lipopolysaccharide can also induce adipose differentiation-related protein expression and promote lipid accumulation in the liver by inhibiting fatty acid oxidation [41]. Lipopolysaccharide acts as a master switch to control adipose tissue metabolism, and its increased levels in the plasma lead to increased adipose tissue differentiation and lipogenesis [40]. Taken together, these findings demonstrate that gut microbiota-derived lipopolysaccharide is sufficient to promote obesity and obesity-related complications. Finally, pro-obesity gut microbiota can significantly contribute to fat accumulation and increase susceptibility to obesity by reducing sensitivity to hormones, such as leptin, or inhibiting the expression of the obesity-suppressing neuropeptides, such as proglucagon [42]. The activities of hormones, such as leptin are critical in maintaining body weight. In the body, leptin helps to regulate and alter long-term food intake and energy expenditure, and it is directly involved in the regulation of body weight. The hormone supports the inhibition of hunger and reduces food intake when the body does not need energy; hence it assists in promoting energy balance. On the other hand, central glucagon-like peptide-1 (GLP-1) is produced in the body and is positively correlated with precursor proglucagon (*Gcg)* mRNA in the nucleus of the brainstem [42]. The expression of GLP-1 has a role in the central regulation of feeding and body fat. Pro-obesity gut microbiota was found to reduce leptin sensitivity and the expression of proglucagon [42]. The decreased proglucagon expression induced by the gut microbiota contributed to the increased fat mass in the experimental mice [42]. These findings provide mechanistic insights into the effect of gut microbiota in promoting obesity and its related disorders. Figure 1 is a diagrammatic representation of the mechanism of action of pro and anti-obesity gut microbiota.

## 3. Gut Microbiota, Obesity, and Hydrolysates/Bioactive Peptides

### 3.1. Significance of Protein Hydrolysates/Peptides in Food and Their Novelty Aspect

Proteins are the primary source of amino acids in the body, and they may also act as energy sources for body metabolic activities when carbohydrate sources are exhausted. In addition to their basic functions in the building and repairing worn-out tissues of the body, proteins also have motifs that may possess bioactivity which contributes to positive health effects [43]. As a result, scientists are currently focusing on the determination of various bioactivities from hydrolyzed protein sources for application as a functional food ingredient. In this regard, there has been an increased investigation on the bioactivity of peptides from hydrolysates of various food-based sources of protein. The food sources of protein may include and are not limited to animal products (milk whey and casein, fish, and egg) and plant sources (wheat and soybean) [6,43]. The protein hydrolysates and bioactive peptides have been reported for their biological functions, including antimicrobial, anti-diabetic, antihypertensive, anti-inflammatory as well as anorexigenic activities [6,43]. There also exist bioactive peptides with dual effects as reported in the literature [6]. Thus, as previously reviewed by other scientists, hydrolysates and bioactive peptides may be promising ingredients for the production of functional foods, especially when combined with other protein products exhibiting properties, such as oil binding, high solubility, emulsifying capacity, and water holding capacity [43]. Production of bioactive peptides involves hydrolysis of food protein to produce hydrolysates and the choice of desired peptides from the end product of hydrolysate (discussed in the subsequent section). Nowadays, mass spectrometry (MS) has enhanced peptide identification. The method has been used to identify peptides due to their high sensitivity, reliability, and speed. Techniques, such as matrix-assisted laser desorption ionization (MALDI) and electrospray ionization (ESI) have been utilized to obtain fragments and ions of peptides that can be distinguished based on the charges in a magnetic field. Sharkey et al. reported that peptides having a proline in their structure at ultimate or penultimate C-terminal position, or on their 1st, 2nd, 3rd, or 4th position from the N-terminal position have generally shown dipeptidyl peptidase 4 (DPP-4) inhibitory activity, the most studied activity of bioactive peptides [43]. Interestingly, DPP-4 has been implicated in many chronic conditions with recent findings showing that the inhibition of DPP-4 activity can substantially reduce obesity and obesity-related inflammation [44] and that pre-treatment with DPP-4 inhibitors in obesity or diabetic condition can protect body organs from these conditions [45]. Hence, it is conceivable that most peptides with anti-obesity activity may have similar characteristics as those displayed by DPP-4 inhibitory peptides. However, this is just a hypothesis, and the exact key structural components of anti-obesity bioactive peptides are not yet fully investigated. Perhaps, this is one area that research has largely ignored. To date, only a few studies are reporting on the molecular structures of peptides with anti-obesity activity.

### 3.2. Production of Protein Hydrolysates and Bioactive Peptides

The production of hydrolysates and peptides was described in the previous reviews [46]. The choice of the method for protein hydrolysis depends on their food origin. Proteins derived from animal products, such as feathers, bristles, or wool contain the keratin structure, hence, they are commonly hydrolyzed using chemicals (acidic or alkaline treatments), or through bacterial keratinases [46]. On the other hand, animal products (including casein, whey, and meat) and plant-based products, such as soy and rice are often hydrolyzed by the use of enzymatic (e.g., cell-free proteases) or microbial treatments. The hydrolysis of proteins results in the production of protein hydrolysates. The protein hydrolysates consist of free amino acids, small peptides, and large peptides [46]. Following hydrolysis, the insoluble fractions are centrifuged or filtered to separate them from protein hydrolysates. Fractionation of protein hydrolysates is then performed to isolate specific peptides or remove undesired peptides which are then purified, characterized, identified, and tested for bioactivity. Even though bioactive peptides can be integrated into foods as a functional ingredient during food processing, they can also be released from food proteins during their transit through the gut. Once released from food along the digestive tract, these compounds are believed to exert activities, such as antioxidant, anti-obesity, anti-inflammatory, and gut modulation effects at a local level [12]. Thus, in the future, we may consume specific foods with proteins that, after digestion and interaction with the microbial enzymes, could generate hydrolysates or peptides with biological activities. Figure 2 describes the steps in the production of anti-obesity food hydrolysates and bioactive peptides.

### 3.3. The Role of Food Hydrolysates/Peptides on Modulating Gut Microbiota in Obesity

In the earlier sections of this review, we described the significance of gut microbiota metabolites in obesity pathogenesis. Different food bioactive components possess great potential in modulating gut microbiota composition and function. Dietary polyphenols [47], protein [48], and food bioactive peptides have been proved to contribute to the composition and functions of gut microbiota. Proteins in foods can be hydrolyzed into fragments, hydrolysates, and/or bioactive peptides. These compounds play important roles in human health and in determining the composition of gut microbial bacteria as well. Recently, evidence from research demonstrates the ability of these food components to change the gut microbiota, which affects body weight. Requena et al. studied the impact of pepsin hydrolysate of egg white (EWH) in obese Zucker rats [11]. The study reported that the daily intake of 750 mg·kg^−1^ EWH in drinking water for 12 weeks modulated the microbiological characteristic by increasing the number of *Lactobacillus/Enterococcus* and *Clostridium leptum* similar to the lean rats than to the obese control. Importantly, Requena et al. discovered that EWH consumption significantly modified the production of gut microbiota metabolites, total short-chain fatty acids in feces [11]. Requena and colleagues hypothesized that the changing of the gut microbiota community occurred mainly due to EWH peptide absorption which modulate of gut microbiota composition [11].

Soybean also contains bioactive peptides or hydrolysates with positive impacts on body weight, and their effects on gut microbiota composition in obesity have been reported. Research investigating the modulatory effects of pepsin-released peptides of soybean 7S globulin on gut microbiota indicated that soybean 7S globulin peptide is involved in lipopolysaccharide-peptide interaction effects of gut microbiota functions related to obesity [49]. The soybean 7S globulin selectively suppressed pro-inflammatory gram-negative bacteria, and also promoted the production of SCFAs due to increased *Lachnospiraceae* and *Lactobacillaceae* relative abundance [49]. Furthermore, a high amount of 7S globulin in soy milk promoted the growth of the *Bacteroides-Prevotella* group in obese men [50], while soybean pepsin-hydrolysate conglycinin maintained a healthy gut microbial community by inhibiting the growth of *E. coli* in mice [51].

*Spirulina platensis*, belonging to the family of *Oscillatoriaceae* are plants that grow naturally in alkaline lakes and seas. *Spirulina platensis* has been used as a food supplement for humans as well as an additive for animal feeds due to its abundant minerals, vitamins, and proteins contents [52]. *Spirulina platensis* protease hydrolysate was investigated for its lipid metabolism and gut microbiota modulation effects in mice [52]. The study found that in addition to its hypolipidemic effect, *Spirulina platensis* protease hydrolysate treatment enriched the abundance of beneficial bacteria in the mice [52]. Following 8 weeks of *Spirulina platensis* protease hydrolysate treatment, the relative abundance of *Alloprevotella, Lachnospiraceae, Prevotella, Ruminococcaceae, Bacteroides, Porphyromonadaceae, Blautia, Desulfovibrionaceae*, and *Porphyromonadaceae* was substantially improved, and *Ruminococcus* composition was the most enriched by the treatment at the genus level [52]. In contrast, the treatment reduced *Allobaculum, Firmicutes, Clostridium_XlVa*, and *Lachnospiracea* [52]. Thus, *spirulina platensis* protease hydrolysate might be used as adjuvant therapy for obesity.

α-Lactalbumin is a major protein component of the whey fraction of bovine milk. The bovine α-lactalbumin hydrolysates can systematically reduce weight gain by modulating the composition of gut microbiota in the obese condition. Bovine α-lactalbumin hydrolysates feeding of rats increased the *Bacteroidetes/Firmicutes* ratio and the relative abundance of *Lachnospiraceae* and *Blautia* [1]. Spearman’s correlation analysis in this study revealed significant correlations between gut microbiota composition and obesity-related indexes [1]. Thus, the bovine α-lactalbumin hydrolysate is a potential functional food ingredient to prevent obesity. Moreover, collagen peptide as a hydrolysate of collagen has been investigated for its beneficial health effects. Collagen peptides have been found to possess health impacts including modifying lipid metabolism [53], anti-aging, and wound healing [54]. As a result of these beneficial health effects, people have been motivated to consume a diet rich in collagen peptides in some areas of the world, such as China. Collagen peptide derived from walleye pollock skin was reported to suppress obesity via modulating gut microbiota in high-fat diet-fed mice [55]. The feeding enhanced the beneficial bacterial count relative to *Lactobacillus, Akkermansia muciniphila, Parabacteroides*, and *Odoribacter spp* while, the intervention reduced the number of intestinal inflammation bacteria, such as *Erysipelatoclostridium* and *Alistipes* [55]. *Akkermansia muciniphila* is well-known for its anti-obesity effects and is also positively correlated with glucose tolerance while *Lactobacillus* is a typical beneficial bacterium in obesity, with studies indicating that the bacterium ameliorates high fat-fed diet-induced blood glucose intolerance and adipose tissue accumulation [56,57]. In a separate study, collagen peptides from two separate food products (*Salmon salar* and *Tilapia nilotica* skins) increased the abundance of *Lactobacillus, Allobaculum*, and *Parasutterella* [54]. The altered gut microbiota was also linked to short chains fatty acids levels, such as butyrate content and shifts in spleen lipid metabolism [54]. Butyrate is a bacterial metabolite known to suppress diet-induced obesity and can also regulate gut hormones via free fatty acid receptors [58]. Thus, collagen peptide from food products is a potential anti-obesity agent which offers an opportunity to develop an adjuvant treatment for obesity. Table 1 summarizes studies on the modulation effects of food hydrolysate/peptide on gut microbiota and the overall impact on obesity parameters.

### 3.4. The Role of Hydrolysates/Peptides in Regulating Obesity-Related Complications via Gut Microbiota Modulation Effects

Oxidation and inflammation are two integral parts of innate immunity and are among the body’s major protective responses to eliminating harmful compounds and signals. When stimulated by metabolic disorders, the bodies’ natural balance is disturbed which may result in the excessive release of radicals and inflammatory factors damaging body cells [62]. Many diseases including cancer, obesity, and neurodegenerative disorders have been correlated with inflammation and oxidative stress. It has been reported that oxidative stress and inflammatory signals concurrently play roles that negatively impact adipose tissue and disrupt the regulation of vascular function in obesity [62]. Research has also revealed that body fluctuations resulting from metabolic disorders contribute to alterations in the gut microbial bacteria balance and disruption of the intestinal epithelial barrier which may trigger the release of pro-inflammatory cytokines and reactive oxygen species [63]. This process eventually leads to a cycle of uncontrolled oxidative stress and inflammation in the body [63]. Thus, activities that reduce chances of developing metabolic disorders, such as weight management can act as appropriate therapeutic approaches to prevent excessive oxidation and inflammatory signaling in the body. Recent studies have demonstrated that some pathogenic gut microbiota through their metabolites have a specific impact on inflammatory responses in the host health. For instance, gut microbiota metabolites, such as lipopolysaccharide can enter into the circulation and bind to the lipopolysaccharide-binding protein in the liver [64]. The complex product formed between lipopolysaccharide and lipopolysaccharide-binding protein further binds to CD14 (a glycolipid-anchored membrane glycoprotein) receptor, which then triggers the activation of several macrophages to produce inflammatory cytokines in the body [64]. Therefore, lipopolysaccharide is one of the important metabolites from pathogenic microbes contributing to inflammation in the body. On the contrary, microbial metabolites, such as butyrate are beneficial microbial metabolites with the ability to preserve the integrity of the intestinal barrier via anti-inflammatory effects. Moreover, some gut microbiota also produce other short-chain fatty acids, which regulate immune responses and eliminate inflammation [65]. Dietary intake restores the balance between beneficial and pathogenic gut microbes and also impacts their roles in the human gut. Only a few data are available explaining how protein hydrolysates/peptides can alter gut microbiota composition which in turn plays a role in preventing obesity-induced inflammation and oxidation. These studies are summarized in Table 2.

Gao and colleagues found out that sturgeon hydrolysate could alleviate inflammation by modifying the *Bacteroidetes*/*Firmicutes* ratio [70]. The treatment by oyster peptide improved the relative abundance of *Alistipes, Lactobacillus, and Rikenell* in mice [67]. *Alistipes* is a member of the *Bacteroidetes* phylum described as a beneficial microbe that could help reduce intestinal inflammation induced by metabolic disorders [71]. Peptide, pyroGlu-Leua also revealed a promising role in the inhibition of inflammatory response via gut bacteria modulation effects in intestinal epithelial cells [67]. According to the study, administration of a pyroGlu-Leu normalized population of *Bacteroidetes* and *Firmicutes* in the colon prevents inflammation of the intestine in mice [67]. These results confirm that hydrolysates/peptide intake may maintain a more balanced composition of gut bacteria and reduce the inflammatory responses from gut bacteria to the host. Therefore, hydrolysates/peptides may be used as prebiotic agents to treat obesity-induced inflammation and oxidations due to their activity in gut microbiota dysbiosis in obese individuals. Figure 3 describes the crosstalk between peptides, gut microbiota, and obesity-induced complications.

### 3.5. Mechanisms of Action of Hydrolysates/Peptides on Gut Microbiota Modulation

Although studies on the relationship between proteins-derived products and gut microbiota have been reported, there is no clear conclusion on the mechanisms of their interaction because of the complexities involved in this relationship. Nevertheless, studies have linked the following events as major mechanisms through which hydrolysates/peptides promote the growth of beneficial gut microbiota. First, it has been reported that hydrolysates/peptides provide essential amino acids which act as natural sources of nitrogen required for the growth of gut microbiota [9]. A study performed on *Lactococcus lactis* demonstrated that the growth of the bacteria was dependent on a nitrogen source from oligopeptides [72]. The pH of the colonic lumen varies with the anatomical site and microbial fermentation of dietary residues [73]. It was revealed that the pH favoring the growth of most colonic bacteria may be approximately 5.5, 6.2, and 6.7 [73]. Since microbial bacteria possess different abilities to use nitrogen from proteins depending on environmental conditions [9], peptides can selectively promote the growth of different gut bacteria at different pH conditions. 

Secondly, hydrolysates/peptides can improve the resistance of strains to acidic environments. As stated above, the osmotic condition of the gut varies between acidic and alkaline. Microbial bacteria along the gut, as a result of fermentation, can also produce organic acids which further alter gut pH homeostasis. Some bacteria are sensitive to pH, and at strong acidic conditions, above the optimum level relative to different bacteria, they can undergo autolysis, leading to a reduced number of viable colonies in the gut [9]. Evidence from Zhang et al. suggests that protein hydrolysates could improve the viability of colonies and preserve their total count by improving their tolerance to extreme acidic conditions [9]. To obtain an insight into this hypothesis, Robitaille et al. conducted a study on the bactericidal effects of acidity on certain strains with or without treatment by caseinomacropeptide or pepsin bovine peptide [74]. The study showed that *Lactobacillus rhamnosus*, a probiotic, treated with caseinomacropeptide or pepsin bovine peptide was more resistant to acid stress compared to untreated strains showing that peptides improve bacterial tolerance to acidic conditions [74]. However, the exact mechanism of acid-resistant induced by peptides on bacterial is not yet fully established. Thirdly, hydrolysates have been reported to enhance the activity of amino peptides and proteases in microorganisms, such as bacteria, thereby, promoting their proliferation [75]. This action may enhance the potential of hydrolysates/peptides to promote the growth of anti-obesity gut microbiota in the body. For instance, hydrolysates from poultry by-products were found to promote the ability of *Lactobacillus* to grow by improving the activities of generic aminopeptidases, such as PepC, PepN, PepL, and PepX [75]. Lastly, some reports have been published showing that hydrolysates/peptides can promote gut bacterial growth by improving their adhesion as reported for human milk and infant formulae hydrolysates on *Bifidobacterium* [76]. 

In contrast, as evident in the previous sections, hydrolysates/peptides can affect the gut microbial balance by reducing the total count of potential pathogenic bacteria in the gut. To the best of our knowledge, there is no specific report in the literature about the mechanism of action of antibacterial effects of hydrolysates/peptides against pathogenic bacteria present in the gut. However, general knowledge about the antimicrobial effects of hydrolysates indicates that peptides, such as those from papain hydrolyzed camel whey produce antimicrobial factors including lysozyme, lactoferrin, and immunoglobulins capable of inhibiting the growth of pathogenic bacteria, such as *E.coli* [77]. Therefore, the hydrolysis of food proteins seems to be a promising tool to produce natural ingredients to alter the composition of gut bacteria that can improve body weight. The mechanisms by which hydrolysates/peptides modulate gut microbiota composition were summarized in Figure 4. 

## 4. Protein Hydrolysates/Bioactive Peptides in the Market Place: Current Issues

The demand for functional ingredients is rapidly growing due to increasing consumer awareness about healthy eating. Bioactive peptides can be incorporated as active ingredients in food at suitable levels that provide health benefits for improved wellbeing. For manufacturers, one of the primary focuses in adding these ingredients to foods is their effects on the organoleptic properties of food products. It has been reported that the addition of protein hydrolysates in foods is hindered due to the presence of low molecular weight peptides consisting mainly of hydrophobic amino acids that may render a bitter taste to foods [79]. Additionally, the addition of protein ingredients in foods containing a high amount of carbohydrates may lead to the formation of Maillard compounds that can influence food flavor or lead to the formation of undesirable compounds [79]. As a result of these negative impacts, the use of these compounds in food has been greatly hindered.

In order to protect consumers against potential health risks, countries have developed a number of regulatory systems that govern the “trade”, “labeling” and “safety” of the bioactive food ingredients with biological effects. Currently, bioactive peptides are sold as functional foods/nutraceuticals/health supplements in the market. Even though at the moment there are no specific regulatory systems for anti-obesity peptides, organizations and countries, such as the European Union (EU), the United States, China, and Canada have already developed a general comprehensive regulatory system for all bioactive peptides or protein hydrolysates sold in the market [80]. In the United States, for instance, bioactivity claims of food derived peptides or protein hydrolysates are classified into two types: structure/function claims and health claims. The structure/function claim classification is related to the benefits of the named peptide on the normal function of the human body. The structure/function claims do not require Food and Drug Administration (FDA) pre-sale approval, as long as the claims are valid without misleading information [80]. Nevertheless, food manufacturers are obliged to write and submit a notification about the claim to the FDA within 30 days after releasing bioactive peptides into the market [80]. Furthermore, such foods with peptides as an additive cannot be sold using phrases, such as “cure”, “treat”, “correct”, “prevent”, or related expressions [80]. Therefore, the market regulation of bioactive peptides or protein hydrolysates is strict and scientific substantiation of the safety of these ingredients is an important aspect that can facilitate faster approval of health claims of these compounds for market release (Table 3 is a summary of the regulatory system in the USA, Canada, European Unio (EU), Japan, and China).

For the proper health function of anti-obesity peptides or other peptides and their incorporation into foods targeting a specific part of the body, the absorption and stability at the target site must be taken into account. The stability and absorption of most of the biologically active compounds in the body is poor. Therefore, encapsulation has emerged as a new strategy to improve the stability and absorption of these compounds in the body. Encapsulation techniques improve not only the stability and absorption of these compounds at the target sites but also their effectiveness. For instance, the peptide PDBSN is a known anti-obesity peptide that can inhibit obesity development by preventing adipocyte differentiation. Nevertheless, the in vivo anti-obesity function has not been certainly pursued due to the poor instability of the peptide during blood circulation [84]. The liposome-encapsulation of peptide PDBSN improved the stability and functionality of the compound at the target site in the body [84]. Currently, encapsulation of bioactive peptides is becoming a novel potential strategy for enhancing the health function of the ingredients. The encapsulation technique can also be used to mask the bitter taste of the peptides, thereby improving their acceptability by the consumers [85]. The final concern emerging with the existence of bioactive peptides in the market is a selection of appropriate packaging materials for finished products. This is a crucial aspect, especially in minimizing the changes in the molecular structure of the peptides during their sale or storage period [86]. Selecting proper packaging protects these compounds from changes, such as those arising due to oxidation [86,87].

## 5. Challenges in the Use of Hydrolysates/Peptides as Anti-Obesity Agents

Protein hydrolysates and peptides have shown biological activity against obesity and obesity-related complications. They have been described as potential ingredients for food, nutraceuticals, or dietary supplements in industries. There are, however, still several challenges that need to be addressed before their widespread utilization as anti-obesity agents: (1) Hydrolysates/bioactive peptides are produced using methods, such as chemical synthesis, enzymatic treatment, and microbial fermentation, which hydrolyzes food protein into smaller forms. Therefore, protein hydrolysates typically are mixtures of different peptides which can be separated based on their sizes, charge, or polarity. Selecting specific peptides with anti-obesity activity from such a mixture is challenging. The lengthy procedure and expenses incurred from hydrolysis, isolation, purification, characterization, identification to the testing stages might be the reason for this challenge. (2) There is still a poor understanding of peptide structure or functional activities because the techniques used to produce well-defined bioactive peptides are relatively small-scale and expensive [88]. (3) It is challenging to estimate and classify the peptide composition of protein hydrolysates making the subsequent identification of the anti-obesity peptides a challenging issue [88]. (4) Most of the studies on weight reduction activities of hydrolysates/bioactive peptides have solely been focused on animal experiments. The lack of human clinical trials showing beneficial effects of food-derived bioactive peptides is a setback in using these compounds. It has become almost impossible to draw a conclusion based on animal experiments on the effectiveness of hydrolysates/peptides as anti-obesity agents in humans. (5) It has been reported that peptides and hydrolysates may negatively impact foods when used as ingredients [88]. When added to food products, peptides may contribute to bad odor, reducing palatability [88]. So, even though they may have potential efficacy against obesity, their use as food ingredients can still be limited. The elimination of these negative impacts on food is still a subject that is yet to be explored. (6) There is a lack of scientific evidence related to absorption, distribution, metabolism, and excretion of bioactive peptides in the body. Thus, there are some fear that consuming large quantities of bioactive peptides may lead to the development of digestive disorders. (7)There are not enough studies on the stability of these compounds during storage or processing time, making their application in foods difficult.

## 6. Perspectives

Studies have established that food hydrolysates promote the growth of beneficial microbes in the gut and suppress the proliferation of pathogenic intestinal microbes and improve body weight; however, the proofs remain inadequate. Although hydrolysates from numerous food products have revealed efficacy in reducing obesity, only a few of them have been investigated for their ability to improve body weight via the gut microbiota modulation effect. Studies are still needed to confirm if hydrolysate/bioactive peptides from other food sources can affect body weight by changing gut microbiota composition and functions. Recently, novel anti-obesity peptides isolated and purified from food proteins have been confirmed to attenuate obesity through other means (such as lipase inhibition, stimulation of digestive hormones, etc.) (Table 4). Additionally, food protein hydrolysates, such as milk whey protein hydrolysates have shown the potential ability to inhibit obesity development via means other than modulating gut microbiota (Table 4).

Investigations are needed to prove if the above novel peptides isolated from foods have the potential to inhibit the development of obesity via modulating gut microbiota composition and functions. Such studies will open the door for the isolation of more peptides from food-rich proteins and their incorporation into commonly consumed foods to target obesity and obesity-related complications. Moreover, efforts are also needed to establish a deeper understanding of the mechanisms by which hydrolysates and peptides promote or inhibit the growth of microbes, and if their activities are influenced by various digestive enzymes present in the gut, something which has only remained partially investigated. Similarly, more studies are needed to uncover areas of application of food hydrolysates/bioactive peptides in the pharmaceutical industries and their incorporation as natural ingredients in functional foods. Finally, while the absorption aspects including passive diffusion, carrier transport, endocytosis have been studied for some hydrolysates/peptides, this knowledge is only limited to hydrolysates/peptides from a limited amount of foods [97]. This information is crucial for understanding the transport pathways of these ingredients and their bioavailability [97]. Therefore, more assessment needs to be incorporated into the future work to understand absorption fate, utilization by gut microbes, and stability of hydrolysates and peptides in the gastrointestinal tract.

## 7. Conclusions

In this review, we attempt to summarize the current knowledge on the interaction between food hydrolysates/peptides and gut bacteria and how this interaction can influence body weight. Food hydrolysates have shown a potential ability to be used as anti-obesity agents. These ingredients can inhibit the growth of pro-obesity gut microbiota but increase the proliferation of anti-obesity gut microbes. These findings suggest that food hydrolysates/bioactive peptides have potential anti-obesity effects and may help combat metabolic disease. Therefore, knowledge of the relationship between food hydrolysates, gut microbes, and obesity is important and can provide a novel strategy to develop food products targeting obesity and related complications.

## Figures and Tables

**Figure 1 antioxidants-11-00333-f001:**
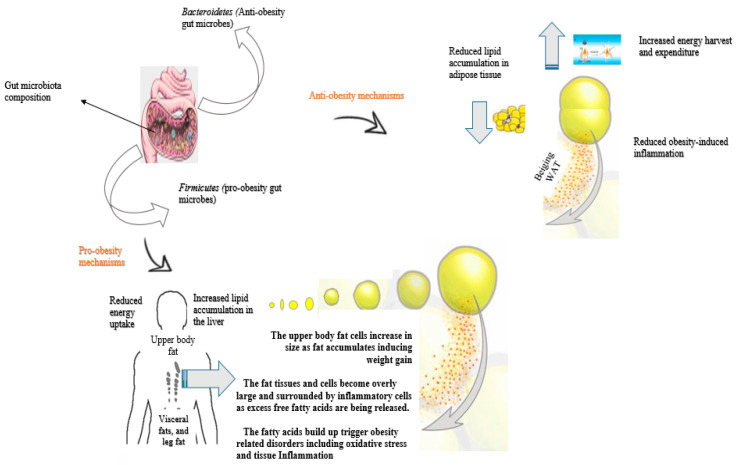
Mechanism of action of pro and anti-obesity gut microbiota. Pro-obesity gut microbiota can induce excessive weight gain by triggering the expression of proteins related to adipose differentiation, increasing liver lipid accumulation, and/or reducing energy uptake in the body. In contrast, anti-obesity gut microbiota reduces risks of obesity by increasing energy expenditure in the body, inhibiting the occurrence of obesity-related inflammation, and/or reducing lipid accumulation in adipose tissue or in the liver. BAT, brown adipose tissue; WAT, white adipose tissue.

**Figure 2 antioxidants-11-00333-f002:**
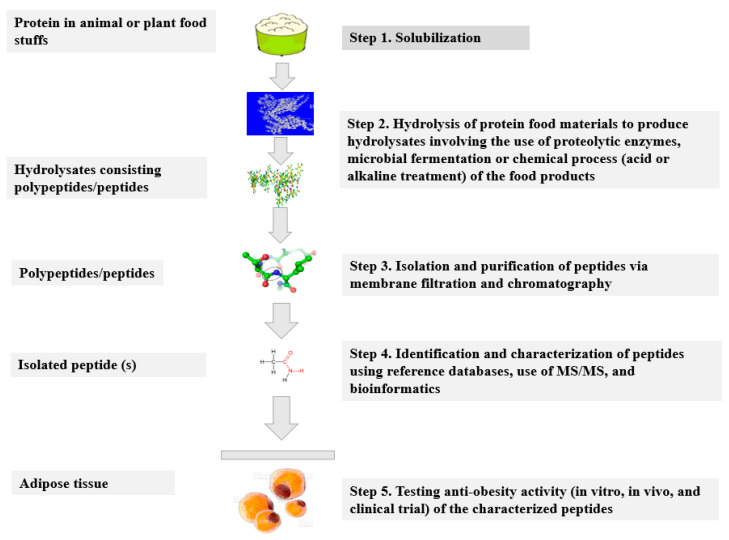
Schematic representation of the production of hydrolysates and bioactive peptides from protein-rich foods. Basically, the hydrolysis of proteins results in the production of protein hydrolysates. The protein hydrolysates consist of free amino acids, small peptides, and large peptides. Following hydrolysis, the insoluble fractions are centrifuged or filtered to separate them from protein hydrolysates. Fractionation of protein hydrolysates follows to isolate specific peptides or remove undesired peptides which are then purified, characterized, identified, and tested for bioactivity.

**Figure 3 antioxidants-11-00333-f003:**
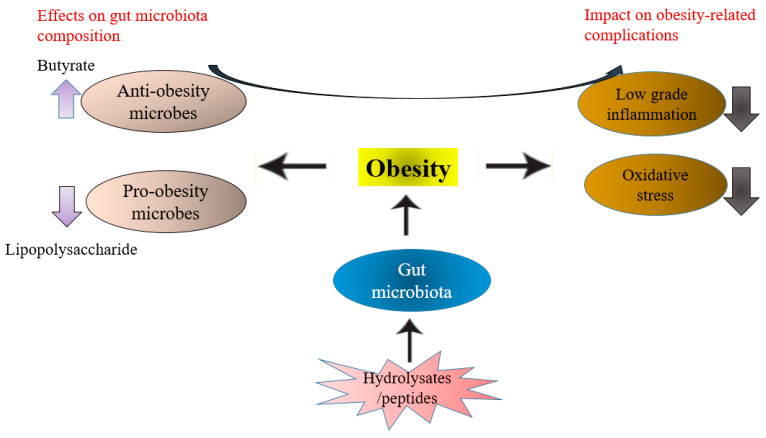
Crosstalk between protein hydrolysates/peptides, gut microbiota, and obesity-induced implications. Obesity can induce low grade inflammation and oxidative stress in an individual by triggering changes in the gut microbiota homeostasis. Protein hydrolysates/peptides can restore gut microbiota dysbiosis. The protein hydrolysates/peptides may act by inhibiting the growth of pathogenic bacteria, such as those producing lipopolysaccharides (which trigger the production of inflammatory cytokines). Similarly, protein hydrolysates/peptides may enhance the proliferation of beneficial gut bacteria. Eventually, the beneficial gut microbiota, through their metabolites, such as butyrate interacts with the host at a molecular level, and this interaction directly or indirectly participates in the inhibition of the onset of oxidative stress and intestinal inflammation in obese individual.

**Figure 4 antioxidants-11-00333-f004:**
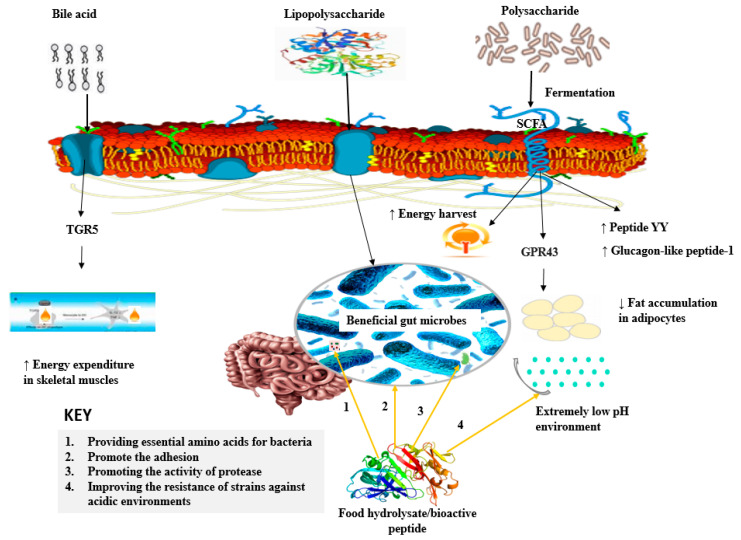
The interplay between food hydrolysates/peptides, gut microbiota, and obesity. Food hydrolysates/peptides improve the growth of beneficial microbes by mechanisms including promoting adhesion, proving essential amino acids, promoting activities of proteases, and improving resistance to an acidic environment. On the other hand, anti-obese gut microbiota residing in the gut can transform de-conjugated primary bile acids into secondary bile acids which thereafter bind to the G-protein-coupled bile acid receptor, Gpbar1 (TGR5), activating energy expenditure in skeletal muscles [78]. Meanwhile, lipopolysaccharides from intestinal epithelial cells may trigger the secretion of pro-inflammatory cytokines which promote obesity complications. Short-chain fatty acids (SCFA), by-products from bacterial fermentation of polysaccharides including butyrate trigger the release of the peptides, such as peptide YY and Glucagon-Like Peptide-1 which suppresses food intake. Further, SCFA-mediated activation of G-protein-coupled receptor 43 (GPR43) suppresses insulin signaling in adipocytes inhibiting fat accumulation in adipose tissue [31].

**Table 1 antioxidants-11-00333-t001:** Modulation effects of food hydrolysate/peptide on gut microbiota and the overall impact on obesity parameters.

Hydrolysate/Peptide	Model	Gut Microbiota-Related Effect	The Overall Impact on Obesity Parameters	Reference
Herring milt hydrolysates	Human	Feeding enhanced *Dubosiella, Lactobacillus*, and *Anaerotruncus*, but reduced *Ruminoclostridum, Flavonifractor*, *Tyzzeralla, Ruminoclostridum, Tyzzerella*, and *Romboustia*	Decreased obesity-related inflammation and Inhibited pro-inflammatory mediators, such as nitric oxide synthase (iNOS)	[59]
Spirulina platensis protease hydrolysate	Mice	Improved *Alloprevotella, Lachnospiraceae*, *Prevotella*, *Ruminococcaceae, Bacteroides*, *Porphyromonadaceae*, *Blautia*, *Desulfovibrionaceae*, and *Porphyromonadaceae* abundance. In contrast, the treatment reduced *Allobaculum*, *Firmicutes*, *Clostridium_XlVa*, and *Lachnospiracea*	Decreased the levels of triglyceride, total cholesterol, low-density lipoprotein cholesterol	[52]
Polysaccharide peptides from Ganoderma lucidum	Mice	Increased the levels of gut microbiota: *Allobaculum, Phascolarctobacterium*, *Psychrobacter, Enterorhabdus, Blautia*, and *Roseburia*. Decreased *Sporosarcina, Bacteroides, Anaerovorax, Parasutterella, Alistipes* and *Alloprevotella*	Decreased serum levels of triglyceride, cholesterol, free fatty acids, and low-density lipoprotein cholesterol (LDL-C)	[60]
Bovine α-lactalbumin hydrolysates	Mice	Increased the *Bacteroidetes/Firmicutes* ratio and the relative abundance of *Lachnospiraceae* and *Blautia*	Alleviated the obesity-related inflammation by reducing expression of transcription factors associated with obesity	[1]
Collagen peptides from Walleye pollock skin	Mice	Enhanced the beneficial bacterial count relative to *Lactobacillus, Akkermansia, Parabacteroides*, and *Odoribacter* spp. Reduced *Erysipelatoclostridium* and *Alistipe*	Reduced serum levels of triglyceride and suppressed the growth of adipocytes and adipose tissue	[55]
Collagen peptide Salmon salar and Tilapia nilotica skins	Mice	Increased abundance of *Lactobacillus, Allobaculum*, and *Parasutterella*, and also increased butyrate production	Decreased pro-inflammatory cytokines, such as TNF-α and lipid metabolism. Upregulated anti-inflammatory (IL-10) cytokines	[54]
Casein Glycomacropeptide Hydrolysates	Mice	Increased the *Bacteroidetes/Firmicutes* ratio, *Ruminiclostridium, Blautia*, and *Allobaculum* abundance	Significantly decreased overall body weight	[61]

**Table 2 antioxidants-11-00333-t002:** Current evidence on the effect of food hydrolysates and peptides on oxidative stress and inflammatory responses via gut microbiota modulation in obese mice.

Hydrolysate/Peptide	Gut Microbiota Effect	Overall Effect	References
PyroGlu-Leu	Normalized population of *Bacteroidetes* and *Firmicutes*	A dose of 0.1 mg/kg induced a significant weight loss and suppressed inflammation of colonic cell	[66]
Oyster peptides	Decreased the proportion of *Firmicutes/Bacteroidetes*, and increased the abundance of *Alistipes, Lactobacillus*, and *Rikenell*	Inhibited the release of inflammatory cytokines	[67]
Pepsin egg white hydrolysate	Increased abundance of *Lactobacillus/Enterococcus* and *Clostridium leptum*	Reduced oxidative stress and inflammation markers. Improved body weight	[11]
Rice endosperm protein derived peptides	Reduced proliferation of pathogenic bacteria, such as *Escherichia coli*	Suppressed endotoxin-related chronic inflammation	[68]
Rapeseed peptide	*Increased Firmicutes* to *Bacteroidetes ratio*	Enhanced the activities of catalase, superoxide dismutase, and glutathione peroxidase enzymes thereby reducing oxidative stress. Decreased overall bodyweight of mice	[69]
Bovine α-lactalbumin hydrolysates	Increased the *Bacteroidetes/Firmicutes* ratio and the relative abundance of *Lachnospiraceae* and *Blautia*.	Decreased the levels of lipopolysaccharide, tumor cell necrosis factor-α, and interleukin-6 in the serum and colon thereby alleviating the obesity-associated inflammation	[1]
Soybean 7S globulin Peptide	Soybean 7S globulin selectively suppressed the growth of pro-inflammatory Gram-negative bacteria.	-	[49]

**Table 3 antioxidants-11-00333-t003:** General regulatory guidelines for bioactive peptides in the USA, Canada, EU, Japan, and China.

Country	Categories of Claim	Existing Regulatory Guidelines	References
USA	Structure/function claim and Health claim	A bioactive peptide sold under a structure/function claim must not use phrases, such as “cure”, “treat”, “prevent”, or related expressions that signify prevention or treatment of a condition (available at https://www.fda.gov/food/food-labeling-nutrition/structurefunction-claims: last accessed on 20 January 2022) The health claims of the bioactive peptides must meet the FDA standard of significant scientific agreement (SSA) and be approved by qualified experts (available at https://www.fda.gov/food/food-labeling-nutrition/authorized-health-claims-meet-significant-scientific-agreement-ssa-standard: last on accessed 20 January 2022) The bioactive peptide must be safe and should display claimed health effect (s) at the required levels without posing any serious problem. All materials involved in the production of bioactive peptides, such as raw material, enzyme, or microbial strain must be GRAS. Any other material must be food-grade and adhere to the federal regulation of the USA. The manufacturers of the compound must obtain a letter of “no objection” to a GRAS notification from the FDA FDA only considers human clinical trials as robust evidence about a health claim. Hence, animal and in vitro studies are insufficient to justify the approval of peptide claim Information, such as those related to intended use, dietary exposure, composition, allergenicity, manufacturing process, stability quality control, and the product specification of the peptides must be well documented.	[80]
Canada	Function claim and Disease reduction claim	The functional claim requires that the biological effects of peptides must not be directly or indirectly associated with the “treatment, mitigation or prevention of any health condition or their symptoms”. The function claim must be specific. For instance, the consumption of 2 mg soy peptide helps to reduce triglycerides levels in the adipose tissue. The health claim must be supported by sufficient evidence from human clinical studies (evidence from animal studies is not sufficient for approval). The dossier must disclose information and guide the proposed daily intake of the peptides. Maximum intake levels of peptides must be stipulated and justified for a target population. Possible side effects and limitations of the consumption of the proposed peptides and possible risk management procedures must be well described.	[80]
European Union	General function claim and Disease reduction claim	Manufacturers should submit an application to EFSA through an EU-country’s competent authority. The EFSAs’ panel under NDA evaluates the scientific evidence of health claims of bioactive peptides and approves it mainly based on scientific substantiation. Humans, animals, and in vitro results must agree in terms of strength, consistency, dose-response relationship, and specificity of the compound. Manufacturers must submit to the EFSA details regarding the characterization of the peptides in terms of molecular weight, amino acid composition, sequences, and length of the peptides, as well as physicochemical properties, conditions of use, and stability of the peptides.	[80,81]
China	Health claim associated with physiological functions	Bioactive peptides with a claim as healthy foods should not be used for disease treatment or as drugs to treat patients. Sufficient scientific evidence from human and animal studies must accompany the health claims of the compound. The testing of bioactive peptides should be conducted by organizations recognized by the China, Food and Drug Administration. The Health Food Expert Committee inspects and approves the applications for approval of bioactive peptides based on the scientific studies and evidence from safety, functionality, stability studies, hygiene inspections as well as and a detailed manufacturing process. The bioactive peptide approved are sold in the market with the Blue Hat logo	[82]
Japan	Foods for Specified Health Use claim and Foods with Nutrient Function Claims	The manufacturer must submit scientific evidence of the efficacy of the bioactive peptide. The evidence must contain results from human studies on the safety of the peptides	[80,83]

FDA, Food and Drug Administration; EFSA, European Food Safety Authority; NDA, Nutrition, Dietetic, and Allergies; GRAS, Generally Recognized As Safe.

**Table 4 antioxidants-11-00333-t004:** Food hydrolysates/peptides that require further investigation on their modulation effect on gut microbiota.

Food Hydrolysate		Effects on Obesity-Related Parameter	Reference
Soluble soy protein peptic hydrolysate	3T3-L1	Up-regulated the expression of peroxisome proliferator-activated receptor γ (PPARγ), a key regulator of adipocyte differentiation	[89]
Protein hydrolysates from β-conglycinin	3T3-L1 and in vitro enzymes test	Downregulated gene expression of lipoprotein lipase (LPL) and fatty acid synthase (FAS). Inhibited nitric oxide synthase (iNOS), a pro-inflammatory mediator	[90]
Milk whey protein hydrolysates	Mice	Significantly reduced plasma total cholesterol levels and body fat content	[91]
Fish protein hydrolysates	STC-1 cells	Stimulated cholecystokinin hormone release	[92]
Novel peptides isolated from food products			
Soy Peptide Phe–Leu–Val	3T3-L1	Reduced Tumour necrosis factor α (TNFα)-induced inflammatory Responses and Insulin Resistance in Adipocytes	[93]
NALKCCHSCPA, NPVWKRK, and CANPHELPNK peptides isolated from Spirulina platensis protein	3T3-L1	Significantly decreased the accumulation of triglyceride	[94]
KDLWDDFKGL and MPSKPPLL from camel milk protein hydrolysate	*In vitro* enzyme analysis	Inhibited porcine pancreatic lipase activities	[95]
RLLPH derived from hazelnut	3T3-L1	Downregulated mRNA expression of adipogenesis-related factors, such as peroxisome proliferator-activated receptor (PPARc), and adenosine monophosphate-activated protein kinase (AMPK) activation	[96]

The table presented food hydrolysates/peptides with known anti-obesity effect through means other than gut microbiota modulation effects. They have been suggested for further investigation on their ability to exert anti-obesity effect via gut microbiota modulation.

## Data Availability

Not applicable.

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
