# Peer review of "The Functional Interplay between Gut Microbiota, Protein Hydrolysates/Bioactive Peptides, and Obesity: A Critical Review on the Study Advances"

_antioxidants, 2022, doi:10.3390/antiox11020333_

Round 1

Reviewer 1 Report

The proposed review is well designed and organized.  It should be published in this form. Because authors did properly explained the role /connection between bioactive peptides, in some cases hydrolysates, microbiota and their role in obesity process. The review gives current concensed knowledge about mentioned above topic.

Author Response

Dear Editor,

RESPONSE TO REVIEWER COMMENTS

We are grateful for your valuable comments and we have carefully revised the manuscript as the reviewers suggested. Please find the response to the reviewer's comments.

Regards,

Deog-Hwan Oh (Ph.D.)

Comment: The proposed review is well designed and organized.  It should be published in this form. Because authors did properly explain the role /connection between bioactive peptides, in some cases hydrolysates, microbiota, and their role in the obesity process. The review gives current concerned knowledge about mentioned above topic.

Response: N/A

ADDITIONAL RESPONSE TO REVIEWERS

Dear reviewer,

In addition to the above comments, the authors wish to bring to your attention the following changes that were made in the original manuscript.

  • The graphical abstract of this article was redrawn for clarity to the readers. Please, refer to the figure below:
  • Figure 1 in the manuscript was modified
  • Table 3 (lines 570-571) was added to the original manuscript. We found the information in table 3 to be interesting and thought it will be helpful to readers. Furthermore, table 3 provides an additional description of section 4: Protein Hydrolysates/Bioactive Peptides in the Market Place: Current Issues

Reviewer 2 Report

This is overall a well-written, thorough review with very nice infographics. I have a few minor comments:

Page 5, lines 170-172. The authors write, “In contrast, Feng et al. showed no relationship between the organism and obesity in their study, but this was influenced by ethnicity, diet, and geographic location of the study subjects.”  Isn’t the influence of microbiota always influenced by ethnicity, diet and geographic location? Please remove that sentence.

Page 5, line 181. The authors write, “Appetite-relater neural signals…” please correct to related.

Page 6, lines 2018-221. The authors write, “The high profile of Bifidobacterium produces a high amount of acetate, which suppresses the growth of pro-obesity gut microbial bacteria such as Escherichia coli and Clostridium perfringens [37]. This interspecific interaction between different intestinal bacteria is crucial in obesity pathogenesis.” It would seem that the association is correlational rather than definitively causal, thus the authors should re-phrase the sentences to reflect that fact.

Page 8, lines 279-280. The authors write, “including antimicrobial, anti-diabetic, antihypertensive, anti-diabetic, anti-inflammatory.” Anti-diabetic is repeated, please omit one.

Several points throughout the paper: the authors write, “inflammations”, please correct this to inflammation.

There have been clinical trials of protein hydrolysates in humans for weight loss; for example, Nobile V, Duclos E, Michelotti A, Bizzaro G, Negro M, Soisson F. Supplementation with a fish protein hydrolysate (Micromesistius poutassou): effects on body weight, body composition, and CCK/GLP-1 secretion. Food Nutr Res. 2016 Jan 29;60:29857. The authors should consider including epidemiological and/or clinical trials in humans in their review.

Author Response

Dear Editor,

RESPONSE TO REVIEWER COMMENTS

We are grateful for your valuable comments and we have carefully revised the manuscript as the reviewers suggested. Please find the response to the reviewer's comments.

Regards,

Deog-Hwan Oh (Ph.D.)

Reviewer 2

  • Comment: Page 5, lines 170-172. The authors write, "In contrast, Feng et al. showed no relationship between the organism and obesity in their study, but this was influenced by ethnicity, diet, and geographic location of the study subjects." Isn't the influence of microbiota always influenced by ethnicity, diet, and geographic location? Please remove that sentence.

Response: The information was deleted

  • Comment: Page 5, line 181. The authors write, “Appetite-relater neural signals…” please correct to related.

Response: The phrase was corrected as ‘Appetite-related neural signals’ (Line 182-183)

  • Comment: Page 6, lines 2018-221. The authors write, “A high amount of acetate, which suppresses the growth of pro-obesity gut microbial bacteria such as Escherichia coli and Clostridium perfringens [37]. This interspecific interaction between different intestinal bacteria is crucial in obesity pathogenesis.” It would seem that the association is correlational rather than definitively causal, thus the authors should re-phrase the sentences to reflect that fact.

Response: This statement was paraphrased as follows “The high profile of Bifidobacterium produces a high amount of acetate, which suppresses the growth of pro-obesity gut microbial bacteria such as Escherichia coli and Clostridium perfringens [37]. Therefore, the characteristic gut microbiota and their metabolites, as well as their relationship, may play a crucial role in the pathogenesis of obesity”

  • Comment: Page 8, lines 279-280. The authors write, “Including antimicrobial, anti-diabetic, antihypertensive, antidiabetic, anti-inflammatory." Anti-diabetic is repeated, please omit one.

Response: The repeated word “Anti-diabetic” was deleted (281-282)

  • Comment: Several points throughout the paper: the authors write, “inflammations”, please correct this to inflammation.

Response: “inflammations” was changed into inflammation in the entire manuscript (Lines 230 and 395)

  • Comment: There have been clinical trials of protein hydrolysates in humans for weight loss; for example, Nobile V, Duclos E, Michelotti A, Bizzaro G, Negro M, Soisson F. Supplementation with a fish protein hydrolysate (Micromesistius poutassou): effects on body weight, body composition, and CCK/GLP-1 secretion. Food Nutr Res. 2016 Jan 29;60:29857. The authors should consider including epidemiological and/or clinical trials in humans in their review.

Response: The review was focused on the interaction between protein hydrolysate/bioactive peptides, gut microbiota, and obesity. Specifically, our goal was to explain how peptides or hydrolysates can modify gut microbiota function/composition to improve body weight. This is the area that we argued has not yet been explored by clinical trials. However, the potential effect of protein hydrolysate/bioactive peptides on obesity via other means, other than gut microbiota modulation has been investigated in clinical trials which include the studies mentioned above.

ADDITIONAL RESPONSE TO REVIEWERS

Dear reviewer,

In addition to the above comments, the authors wish to bring to your attention the following changes that were made in the original manuscript.

  • The graphical abstract of this article was redrawn for clarity to the readers. Please, refer to the figure below:
  • Figure 1 in the manuscript was modified
  • Table 3 (lines 570-571) was added to the original manuscript. We found the information in table 3 to be interesting and thought it will be helpful to readers. Furthermore, table 3 provides an additional description of section 4: Protein Hydrolysates/Bioactive Peptides in the Market Place: Current Issues
